# Molecular Profiling of Axial Spondyloarthritis Patients Reveals an Association between Innate and Adaptive Cell Populations and Therapeutic Response to Tumor Necrosis Factor Inhibitors

**DOI:** 10.3390/biom14030382

**Published:** 2024-03-21

**Authors:** Daniel Sobral, Ana Filipa Fernandes, Miguel Bernardes, Patrícia Pinto, Helena Santos, João Lagoas-Gomes, José Tavares-Costa, José A. P. Silva, João Madruga Dias, Alexandra Bernardo, Jean-Charles Gaillard, Jean Armengaud, Vladimir Benes, Lúcia Domingues, Sara Maia, Jaime C. Branco, Ana Varela Coelho, Fernando M. Pimentel-Santos

**Affiliations:** 1Applied Molecular Biosciences Unit, Life Sciences Department, Sciences and Technology School, NOVA University of Lisbon, 2829-516 Caparica, Portugal; 2Instituto de Tecnologia Química e Biológica António Xavier, Nova University of Lisbon, Av. Da República, 2780-157 Oeiras, Portugal; ana.filipa.rfernandes.96@gmail.com (A.F.F.); varela@itqb.unl.pt (A.V.C.); 3Department of Medicine, Faculty of Medicine, University of Porto, 4099-002 Porto, Portugal; u003616@chsj.min-saude.pt; 4Rheumatology Department, Centro Hospitalar e Universitário de São João, 4200–319 Porto, Portugal; 5Rheumatology Department, Centro Hospitalar de Vila Nova de Gaia/Espinho, 4434-502 Vila Nova de Gaia, Portugal; 6NOVA Medical Research, NOVA Medical School, NOVA University of Lisbon, 1169-056 Lisbon, Portugal; helena.santos@ipr.pt (H.S.);; 7Portuguese Institute of Rheumatology, 1050-034 Lisbon, Portugal; 8Rheumatology Department, Centro Hospitalar do Tâmega e Sousa, Hospital Padre Américo, 4560-136 Penafiel, Portugal; 74479@chts.min-saude.pt; 9Rheumatology Department, Unidade Local de Saúde do Alto Minho, 4990-078 Ponte de Lima, Portugal; 10I.CBR—Institute for Clinical and Biomedical Research, Faculty of Medicine, University of Coimbra, 3000-548 Coimbra, Portugal; 11Rheumatology Department, Centro Hospitalar e Universitário de Coimbra, 3004-561 Coimbra, Portugal; 12Rheumatology Department, Centro Hospitalar Médio Tejo, 2350-754 Torres Novas, Portugal; 13Département Médicaments et Technologies pour la Santé (DMTS), Université Paris-Saclay, CEA, INRAE, SPI, F-30200 Bagnols-sur-Cèze, France; 14EMBL Genomics Core Facility, Meyerhofstr. 1, D-69117 Heidelberg, Germany; benes@embl.de; 15Escola Superior de Saúde, Instituto Politécnico de Setúbal, 2910-761 Setúbal, Portugal; 16Rheumatology Department, Centro Hospitalar de Lisboa Ocidental, Hospital Egas-Moniz, 1349-019 Lisbon, Portugal

**Keywords:** axial spondyloarthritis, TNF inhibitor (adalimumab), treatment response, disease activity, innate immune system, adaptive immune system, peripheral blood, RNA-seq, proteomics

## Abstract

This study aims at identifying molecular biomarkers differentiating responders and non-responders to treatment with Tumor Necrosis Factor inhibitors (TNFi) among patients with axial spondyloarthritis (axSpA). Whole blood mRNA and plasma proteins were measured in a cohort of biologic-naïve axSpA patients (*n* = 35), pre and post (14 weeks) TNFi treatment with adalimumab. Differential expression analysis was used to identify the most enriched pathways and in predictive models to distinguish responses to TNFi. A treatment-associated signature suggests a reduction in inflammatory activity. We found transcripts and proteins robustly differentially expressed between baseline and week 14 in responders. C-reactive protein (CRP) and Haptoglobin (HP) proteins showed strong and early decrease in the plasma of axSpA patients, while a cluster of apolipoproteins (APOD, APOA2, APOA1) showed increased expression at week 14. Responders to TNFi treatment present higher levels of markers of innate immunity at baseline, and lower levels of adaptive immunity markers, particularly B-cells. A logistic regression model incorporating ASDAS-CRP, gender, and *AFF3*, the top differentially expressed gene at baseline, enabled an accurate prediction of response to adalimumab in our cohort (AUC = 0.97). In conclusion, innate and adaptive immune cell type composition at baseline may be a major contributor to response to adalimumab in axSpA patients. A model including clinical and gene expression variables should also be considered.

## 1. Introduction

Axial Spondyloarthritis (axSpA) can lead to significant disability and impairment in quality of life [1]. Clinical features of axSpA are heterogeneous, including inflammatory back pain, asymmetrical peripheral oligoarthritis (predominantly of the lower limbs), and enthesitis, and specific organ involvement, such as acute anterior uveitis, psoriasis, and chronic inflammatory bowel disease [2].

In axSpA, non-steroidal anti-inflammatory drugs (NSAIDs) have a central role in treatment and are considered the first choice therapy. However, biological disease-modifying antirheumatic drugs (bDMARDs), including Tumor Necrosis Factor inhibitors (TNFi), Interleukin-17 and Interleukin-23 inhibitors (IL-17i, IL23i), and, more recently, target synthetic DMARDs (tsDMARDs) as Janus Kinase (JAK) inhibitors, are recommended to patients enduring active disease despite conventional treatment (or intolerance/contraindication) [3]. The efficacy of bDMARDs has been documented in several studies showing significant and early improvements in disease activity and function [4,5] sustainable for long periods of time [6,7,8]. In spite of its well documented benefit in axSpA, up to 40% of patients fail to respond to bDMARD treatment [9] or suffer adverse events [10,11].

Patients that fail to respond to the first bDMARD usually switch to another (with the same or another mechanism of action), and it may take several iterations to find a suitable drug that reduces disease activity effectively [3]. Response to an effective therapy can take several months, and the delay for non-responders implies continued impact of disease, and, potentially, additional irreversible damage and potential exposition to adverse events. In this context, it is important to identify, as early as possible, patients highly (un)likely to respond to therapy, following the treat-to-target concept [12] and in line with the concept of “window of opportunity” [13,14].

Studies specifically in axSpA indicate that primary non-responders to TNFi tend to be older, *HLA-B27* negative, have higher baseline structural damage, and poor function, Ref. [15] and are treated with soluble TNF receptors [9]. Likewise, some markers were identified in association with a good response to therapy, namely: younger age, *HLA-B27* carriage, elevation of acute phase reactants (CRP), and marked spinal inflammation, as evaluated by MRI [16].

No studies so far have identified molecular changes associated with a good/bad response to TNFi treatment in axSpA. However, several studies have tried to develop molecular predictors of response to TNFi treatment in rheumatoid arthritis (RA), with variable success [17,18,19]. One such study using whole blood transcriptome achieved 65% accuracy in predicting response to infliximab using a 10-gene biomarker set [17]. In another study using transcriptome of monocytes, *CD11C* was found to be a very good predictor of response (95% accuracy) to adalimumab monotherapy in RA [18]. Thomson and colleagues used publicly available data to develop a model that increased the capacity to detect non-responders, from 27% to 59%, using 18 signaling mechanisms [19]. More recently, a set of studies suggested an interplay between innate and adaptive immunity, with a higher myeloid-driven inflammation in responders and higher lymphoid activity in non-responders [19,20,21].

Reliable predictors of outcome for TNFi monotherapies in axSpA are not yet available. The goal of this study is to identify predictors, at baseline, of patient response to TNFi therapy (adalimumab) in axSpA using transcriptome and proteome approaches in peripheral blood samples.

## 2. Materials and Methods

### 2.1. Study Design and Samples Collection

The Bioefficacy study, *Biomarkers Identification of Anti-TNFα Agent’s Efficacy in Ankylosing Spondylitis Patients Using a Transcriptome Analysis and Mass Spectrometry* (clinical trials.gov identifier NCT02492217), is multicentric and conducted across multiple Rheumatology departments in mainland Portugal. It is a prospective, non-randomized study, spanning 14-weeks focusing on adult patients with axSpA (detailed in Table 1), aged between 18 and 75 years old, and fulfilling the axSpA ASAS criteria [22]. The study was conducted from 2014 to 2019, and involved biologic-naïve patients initiating TNFi therapy with adalimumab (40 mg subcutaneously fortnightly), according to the Portuguese Society of Rheumatology Guidelines [23] (see Appendix A). The detailed study protocol has been published. Clinical evaluations (Bath Ankylosing Spondylitis Activity Index—BASDAI, Bath Ankylosing Spondylitis Functional Index—BASFI, Bath Ankylosing Spondylitis Metrology Index—BASMI, and 36-Item Short-Form Health Survey—SF-36) and peripheral blood collections were performed at baseline (BL) (start bDMARD) and after 3–5 days (D3), 2 weeks (W2), and at 14 weeks (W14). Patients were classified as responders or non-responders, according to ASAS20 [24,25] at week 14. To increase confidence in the response assessment, ΔASDAS and BASDAI 50 [16] were used as secondary response criteria. All clinical evaluations were performed by previously trained rheumatologists. Blood samples were collected from all subjects at baseline to test for *HLA-B27* status and at each timepoint to determine C-reactive protein (CRP), Erythrocyte Sedimentation Rate (ESR), and other biochemical parameters, and for RNA-seq and serum proteome analysis (further details in Appendix A).

### 2.2. Data Analysis

Descriptive statistics were used to summarize baseline characteristics for responders and non-responders. Two sample Wilcoxon tests (continuous variables) and chi-square tests of association (categorical variables) were used to compare baseline characteristics between responders and non-responders.

Differential gene and protein expression analysis used the limma R package (R version 4.1, Bioconductor 3.14) to apply a voom transformation for variance stabilization on normalized expression values, and to obtain differentially expressed genes through an empirical Bayes method, followed by multiple test correction with the Benjamini–Hochberg method. Genes and proteins were considered differentially expressed if the adjusted *p*-value of the test was less than 0.05.

Logistic regression models and plotting were performed using the R software as above. Sparse partial least squares discriminant analysis (sPLS-DA) was performed using the mixOmics R package. Random forest models were obtained using the randomForest R package.

Further details can be found in Appendix A.

## 3. Results

### 3.1. TNFi (Adalimumab) Leads to a Decrease in Disease Activity in the Majority of axSpA Patients

Of the 58 patients enrolled in the entire study (40 responders and 18 non-responders), 36 patients (the 18 responders with the best response and all the 18 non-responders) were selected for transcriptome and proteome analysis (1 non-responder was later excluded due to the unavailability of high-quality biological samples). Table 1 briefly summarizes the clinical characteristics of this cohort.

At BL, responders exhibited higher levels of C-reactive protein (CRP) (*p* = 0.011) and ASDAS-CRP (*p* < 0.001). Additionally, responders also have a higher proportion of HLA-B27 positivity (*p* = 0.01). Disease activity showed a decrease from BL to W14 in both responders (mean ∆ASDAS-CRP: 2.6, *p* < 0.001; mean ∆BASDAI: 4.6, *p* < 0.001) and non-responders (mean ∆ASDAS-CRP: 0.7, *p* < 0.001; mean ∆BASDAI: 1.3, *p* = 0.006) (Appendix A). Other clinical attributes were comparable between the two groups.

This suggests, as expected, that treatment with adalimumab, with a few exceptions, has lowered inflammatory markers and disease activity scores in most patients.

### 3.2. Treatment with Adalimumab Had a Significant Impact on the Expression of Blood Cell Transcripts and Plasma Proteins of axSpa Patients

In an unsupervised principal component analysis (PCA), the expression levels of blood cell transcripts and abundances of plasma proteins in axSpA patients did not clearly separate responders from non-responders, at neither BL nor W14 (Figure 1A). Nonetheless, plasma proteomics showed clear differences between BL and W14 in responders, suggesting an effective impact of adalimumab treatment. Indeed, a sparse partial least squares discriminant analysis (sPLS-DA) supports a separation between BL and W14, for both responders and non-responders, not only in proteomics (Appendix A), but also in transcriptomics (*p* < 0.05, Figure 1B).

Permutational multivariate analysis of variance indicates that both time point (3% and 17%) and response group (2% and 4%) can explain a small but statistically significant (*p* < 0.05) part of the observed global variation in both transcript and protein levels, respectively. Moreover, sPLS-DA analysis supports a separation between responders and non-responders at baseline (*p* < 0.01, Figure 1C).

This suggests that treatment with TNFi had a significant impact in the expression of blood cell transcripts and plasma proteins of axSpA patients undergoing treatment with adalimumab. Moreover, it also suggests the existence of detectable differences between responders and non-responders at baseline.

### 3.3. Transcripts and Proteins Varying between Baseline and Week 14 Were Associated with a Decrease in Innate Immune Activity

In responders, 2120 (of 21438) genes (103 genes with fold change (FC) greater than 2) and 41 (of 129) proteins (7 with FC > 2) were differentially abundant between BL and W14, of which 1096 genes (41 with FC > 2) and 25 proteins (4 with FC > 2) were upregulated at W14 (Figure 2A, Appendix A). Genes associated with inflammation, particularly neutrophil-driven (such as *DOK3*, *LRG1*, and *MMP9*), tended to be significantly less expressed in blood cells at W14 in comparison to BL, while upregulated genes were associated mostly with translation and other metabolic processes (e.g., *EEF1A1*, *RPL7*, *MRPL1*, Figure 2B). In agreement with this, plasma proteins less abundant at W14 were associated with the activation of the complement system and innate immunity, including the complement factors CFB and CFH and complement components C3, C8B, and C8G (Figure 2B, Appendix A). Plasma proteins more abundant at W14 were linked with vitamin metabolism, including the apolipoproteins APOA1, APOA2, and APOA4. Given the consistent decrease in expression of neutrophil and innate immunity markers, we also compared estimated frequencies of different white blood cells between BL and W14. In agreement with the gene expression results, we observed in responders a significant decrease in neutrophil frequency at week 14 and an increase of B cell frequency (Figure 2C), with a similar pattern observed for other adaptive immune cell populations such as CD4+ T-cells (Appendix A).

In non-responders, no significant differences in blood cell gene expression were identified between BL and W14. However, a rank-based gene set enrichment analysis (GSEA) of the transcriptome data revealed the same pathways as those observed in responders (Appendix A). Sixteen plasma proteins were found to be differentially expressed (none with FC > 2), with eleven of them upregulated at W14. Notable examples include Apolipoprotein A1 (APOA1), C-Type Lectin Domain Family 3 Member B (CLEC3B), Complement Factor H (CFH), and Retinol Binding Protein 4 (RBP4) (Appendix A). Based on proteomic data, no pathways were found significantly different regulated. Also, although there is a similar tendency to decrease neutrophil frequency between BL and W14 in non-responders, it does not reach statistical significance in neutrophils or other immune cell populations (Appendix A).

The levels of transcripts and proteins that were differentially abundant between BL and W14 in responders were more similar at W14 (*p* < 0.05) than they were at BL, suggesting preexisting differences at baseline that were attenuated due to treatment (Appendix A). In agreement with this observation, we did not find any genes or proteins displaying significantly different behavior between time and response group, suggesting that the action of adalimumab in responders and non-responders is similar.

Overall, these results suggest that the transcripts and proteins that varied during adalimumab treatment were associated with a decrease in innate immune activity.

### 3.4. Markers of Inflammation Exhibited a Decrease in Plasma Levels as Early as 3 Days after Adalimumab Treatment, Observed in Both Responders and Non-Responders

To refine our understanding of the temporal response to adalimumab, we also performed plasma proteomics analysis at 3–5 days (D3) and 2 weeks (W2) after beginning of treatment.

In responders, several plasma proteins that were significantly downregulated at W14 compared to BL, including Haptoglobin (HP), Haptoglobin receptor (HPR), and CRP, exhibited a tendency to decrease already at D3 (*p* = 0.07, 0.2, and 0.02, respectively), with further reduction until W2 (*p* < 0.001 for all; Figure 3A, Appendix A). In non-responders these proteins also showed a similar decreasing trend at D3 and W2, although not reaching statistical significance (Appendix A).

In responders, among the proteins significantly increasing at W14 compared to BL, there was greater heterogeneity. However, some proteins, such as ApolipoproteinD (APOD), Apolipoprotein A2 (APOA2), and a recombinant protein of human pro-platelet basic protein (chemokine (C-X-C motif) ligand 7) (PPBP), displayed a tendency to increase their abundance already at W2 (*p* = 0.3, 0.01, and 0.03). Interestingly, the average level of change of these upregulated proteins was much milder (maximum FC of 2) when compared to the downregulated proteins HP, HPR, and CRP (FC of 3–5). A similar trend of increased abundance at W2 of APOD, APOA2, and PPBP was observed in non-responders (*p* = 0.002, <0.001, 0.1; Figure 3B, Appendix A).

Thus, our results indicate that some markers of inflammation elevated at baseline were already lowered in the plasma of some patients after just 3–5 days of adalimumab treatment, with similar patterns observed in both responders and non-responders.

### 3.5. Blood Transcriptome Data at Baseline Suggest That Response to Adalimumab Stems from an Interplay between Innate and Adaptive Immunity

At BL, no proteins (of 112) were found to be significantly differentially abundant between responders and non-responders, but we could identify 92 genes (of 18,688, 12 with FC > 2) that were differentially expressed between responders and non-responders. Among these, 16 (0 with FC > 2) were more expressed in responders, while 76 (12 with FC > 2) were more expressed in non-responders (Figure 4A, Appendix A). Genes with higher expression in responders were associated with inflammation, including processes such as neutrophil degranulation and interferon signaling. On the other hand, genes more expressed in non-responders were linked to lymphocyte activation, particularly B-cell activity, and metabolism, specifically translation (Figure 4B). Notably, among the top differentially expressed genes were *PAX5*, *CD20* (*MS4A1*), *FCRLA*, and *BANK1*, all associated with B-cell activity and all significantly more expressed in non-responders at BL (Figure 4C).

Supporting this observation, the estimation of lymphocyte population frequencies using RNA-Seq indicated significantly higher frequencies of B-cells in non-responders at BL (Figure 4D). Genes associated with B-cells exhibited the most pronounced overall difference between responders and non-responders at BL, and between BL and W14 in responders (Appendix A), with a similar pattern to that observed for other adaptive immune cell populations. In contrast, neutrophil-associated genes showed an opposing pattern, being significantly downregulated between BL and W14. Moreover, a significant positive correlation was found between disease activity and neutrophil frequencies, while a negative correlation was observed with B-cells and T-cells (Appendix A).

Thus, our results suggest that response to adalimumab derives from alterations in the balance between innate and adaptive immunity, indicating an opposing role, particularly between neutrophils and B-cells.

### 3.6. The Blood Transcriptome Data Enhance the Ability to Differentiate between Responders and Non-Responders at Baseline

In our cohort, ASDAS-CRP at BL showed a significant association with TNFi response in a multivariate logistic model (Figure 5A). HLA-B27 status approached a borderline *p*-value of 0.07, while biological gender, age at diagnosis, and disease duration did not reach statistical significance. Responders exhibited a higher ASDAS-CRP, with an optimal threshold of 4.15 (100% sensitivity and 50% specificity), identified using the ROC curve (area under the curve (AUC) = 0.83, Figure 5B). A model that incorporated ASDAS-CRP, gender, and HLA-B27 achieved an AUC of 0.9. Notably, a model replacing HLA-B27 with the ratio between neutrophils and lymphocytes reached an AUC of 0.87.

The gene *AFF3*, a tissue-restricted nuclear transcriptional activator preferentially expressed in lymphoid tissue, emerged as the top differentially expressed gene between responders and non-responders at BL. Integrating the gene expression values of *AFF3* into a logistic regression model, along with gender and ASDAS-CRP, elevated the AUC to 0.97 (Figure 5B). Additionally, a random forest model utilizing the same variables achieved a predicted accuracy of 80–85%, surpassing the performance when using ASDAS-CRP alone (60%) or when relying solely on clinical variables (70–75%).

Our results suggest that blood transcriptome data can improve our ability to differentiate responders from non-responders at baseline, and that simple hemogram data may have valuable clinical application. 

## 4. Discussion

Our findings indicate that TNFi leads to a reduction in inflammatory markers in the majority of axSpA patients, aligning with observations from prior studies [4]. Both our transcriptome and proteome data consistently suggest a global decrease in inflammatory markers 14 weeks after initiating TNFi, corroborating changes in clinical markers of inflammation and disease activity scores. Specifically, genes showing reduced expression between BL and W14 of treatment are linked to inflammation and immune-related pathways. The proteome analysis further supports this observation, indicating a decreased expression of proteins associated with inflammation, innate immunity, and the complement system. In contrast, genes exhibiting increased expression at W14 are associated with metabolic processes, particularly gene translation and lipid metabolism.

Our transcriptome results closely align with previous findings in axSpA: *TNFSF14* (*LIGHT*), *IL17RA*, *EPOR*, and *IFNAR1*, the genes highlighted by Haroon et al. [26], exhibited significantly reduced expression after TNFi treatment in our cohort. Moreover, 58 (16%) of 360 genes upregulated after TNFi in the study by Wang et al. [27] were also upregulated in our investigation, and 88 (31%) of the 285 downregulated genes were similarly identified in our study. This suggests that, despite the heterogeneity in clinical manifestations of axSpA, the molecular response to TNFi appears consistent across different studies, at least at the level of blood cell transcriptome.

In addition, among the 103 significantly differentially expressed genes (DEGs) between BL and W14 with higher fold changes (FC > 2), 10 had been previously identified as associated with axSpA in genome-wide association studies (GWAS) [28]. Of these, *TNFRSF1A*, *TBKBP1*, *HHAT*, and *LTBR*, all exhibiting lower expression at W14, are involved in the TNF pathway, mediating apoptosis through nuclear factor-κB [29], and serving as regulators of inflammation. Notably, *IL1R*, *IL6R*, and *TYK2*, more associated with innate immunity, were downregulated, while *IL7R* and *ICOSLG*, more associated with the adaptive immune system, particularly the stimulation and differentiation of T- and B-cells, were upregulated. *FCGR2A*, also downregulated, encodes a cell surface receptor found on phagocytic cells such as macrophages and neutrophils [28]. We opted for *AFF3* for our model as the most differentially expressed gene in our cohort, even though it was not identified in previous GWAS analyses. Future studies should aim to elucidate the physiopathologic role of these genes in the response to TNFi.

We also have identified several plasma proteins that undergo significant changes in abundance with TNFi treatment. In general, we observed a decrease in the abundances of HP, HPR, CRP, and complement factors, along with an increase in several apolipoproteins, CLEC3b, and RBP4. Notably, in responders, we observed an early (since days 3–5) and persistent decrease in HP and CRP, which correlated with observed clinical improvement. The decrease in complement factors was milder and occurred later (after W2). The subsequent increase in apolipoproteins APOA1, APOA2, and APOD, known for their involvement in lipid clearance from circulation and anti-inflammatory properties, has been demonstrated in previous studies [30].

Interestingly, APOD, which lacks marked similarity to other apolipoprotein sequences, shows a high degree of homology to plasma retinol-binding protein (Rbp), which is also overexpressed. Both proteins are believed to influence bone metabolism, with RBP4 found in a restricted population of epiphyseal chondrocytes and perichondral cells, likely correlating with future regions of secondary ossification [31]. Additionally, CLEC3B, a plasminogen-binding protein induced during the mineralization phase of osteogenesis, is also more abundant [32]. These findings shed light on the molecular mechanisms underlying the resolution of inflammation while also suggesting that osteoproliferation may be induced under TNFi therapy, as documented in recent studies [33,34,35].

The primary objective of our study was to identify molecular predictors, at baseline (BL), of the response to TNFi. Our findings support ASDAS-CRP as an effective measure for promptly determining TNFi as a therapeutic option in axSpA. However, for cases with moderate disease activity, additional variables are necessary for accurate prediction. The inclusion of *AFF3*, the top differentially expressed gene at BL between responders and non-responders, in a logistic regression model with ASDAS-CRP and gender enabled precise prediction of the response to adalimumab in a high proportion of patients (AUC = 0.97). Moreover, robust machine learning methods, including cross-validation, suggest a predictive capacity exceeding 80% accuracy.

The identification of an interplay between innate and adaptive immunity, consistent with findings in previous studies in rheumatoid arthritis (RA) [21], implies similar mechanisms in both diseases. Our analysis suggests that ratios between innate and adaptive immune populations, such as neutrophil/lymphocyte ratios, merit further exploration as a clinical marker of interest for TNFi therapeutic decisions.

To our knowledge, this study is the first to utilize a multi-omic approach to address the challenging task of predicting therapeutic response to TNFi in the context of axSpA, involving a significant number of participants followed for 14 weeks, which is the minimum time required to evaluate response in clinical practice. In addition, this work contributes to validate results from previous studies. Nonetheless, we acknowledge limitations in our study, such as the involvement of patients from only one country, the use of only one TNFi, and the short period of follow up.

In conclusion, we have documented a significant impact of adalimumab treatment on transcript expression and protein abundances during the initial 14 weeks of treatment. Our results suggest an interplay between innate and adaptive immunity under TNFi therapy, with lymphoid markers emerging as the most differentially expressed between groups and enabling highly accurate predictive models within our cohort. Taken together, our findings indicate that molecular data can not only provide mechanistic insights into the genesis and progression of the disease, but also suggest novel biomarkers for evaluating the potential response to adalimumab and probably other TNFi before initiating treatment.

## Figures and Tables

**Figure 1 biomolecules-14-00382-f001:**
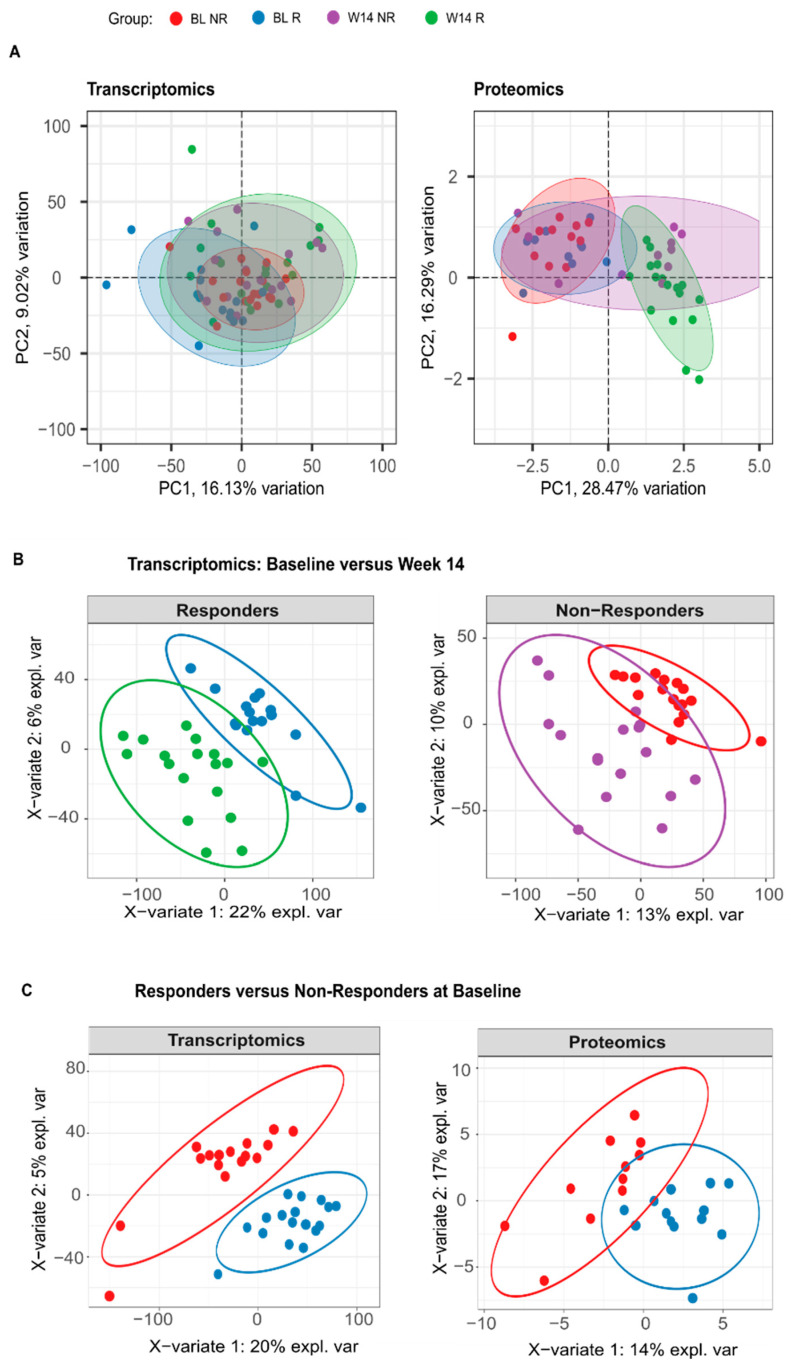
Response to TNFi has a significant impact on the relative abundance of blood cells transcripts and serum proteins of patients. (**A**) Principal component analysis (PCA) of the blood cell transcriptomics and proteomics data for responders (R) and non-responders (NR) at baseline (BL) and week 14 (W14). For visual clarity two outliers are out of view in the transcriptomics PCA, but all data were used to generate the plot. (**B**) Sparse partial least squares discriminant analysis (sPLS-DA) of transcriptomics data, using time as a variable of interest, in responders (AUC = 0.99, permutation test *p* < 0.001) and non-responders (AUC = 1, *p* < 0.001). (**C**) The sPLS-DA of transcriptomics (AUC = 1, *p* < 0.001) and proteomics (AUC = 1, *p* < 0.001) data at baseline, using response group as a variable of interest. In all cases, AUC and *p*-value correspond to the two best components of the sPLS-DA. In all graphs, ellipses represent 95% confidence intervals.

**Figure 2 biomolecules-14-00382-f002:**
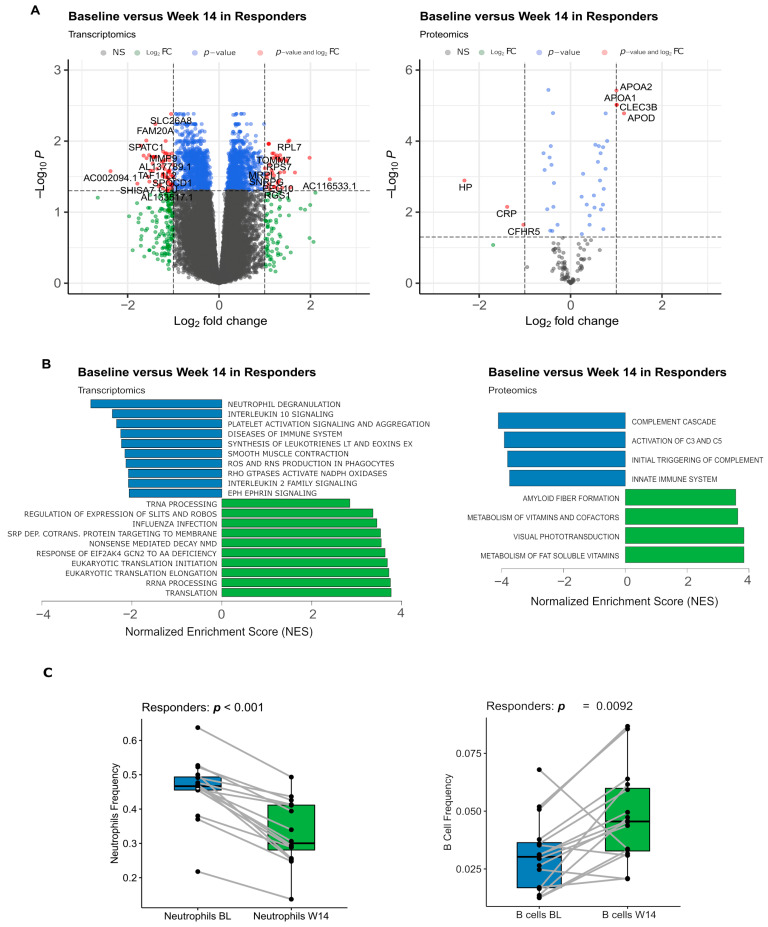
Response to TNFi has a significant impact in the relative abundance of the blood cell transcripts and plasma proteins of patients. (**A**) Volcano plots (log2 of the fold change versus −log10 of the false discovery rate (FDR)) comparing the levels of blood cell transcripts and plasma proteins in baseline samples versus week 14 samples in responders. In grey are non-significant (NS) genes/proteins; in green are genes/proteins that are not statistically significant (FDR > 0.05) but have an estimated fold change greater than 2; in blue are genes/proteins that are statistically significant but have a milder fold change (less than 2); and in red are genes/proteins that are statistically significant and have a fold change greater than 2. The identifiers of all the red proteins and some of the red genes are displayed in the plot. (**B**) Barplots displaying the Normalized Enrichment Score (NES) of representative significant pathways resulting from a gene set enrichment analysis (GSEA) comparing the gene expression or protein abundances of W14 (green) against BL (blue) responder samples. (**C**) Boxplot displaying estimated relative frequencies of neutrophil and naive B-cell in BL and W14 samples. In C, the *p*-value is from a paired Wilcoxon rank-sum test; samples of the same patient are connected with a grey line.

**Figure 3 biomolecules-14-00382-f003:**
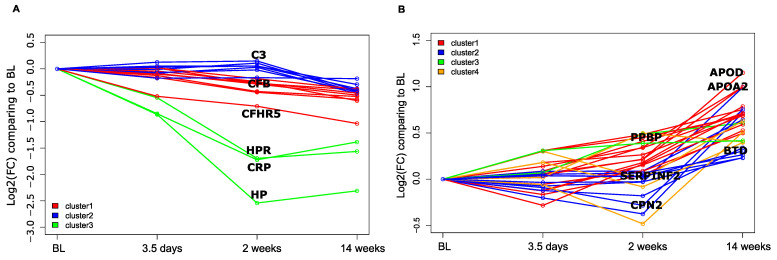
Markers of inflammation are already lowered in the plasma after 3–5 days of adalimumab treatment in both responders and non-responders. (**A**) Log2 fold change of proteins between a given time point and the baseline. Only proteins significantly downregulated at W14 in responders were represented. (**B**) Same as (**A**) but with upregulated proteins. Proteins with similar temporal behavior were clustered using the dtwclust R package. Only the names of a set of representative proteins are displayed.

**Figure 4 biomolecules-14-00382-f004:**
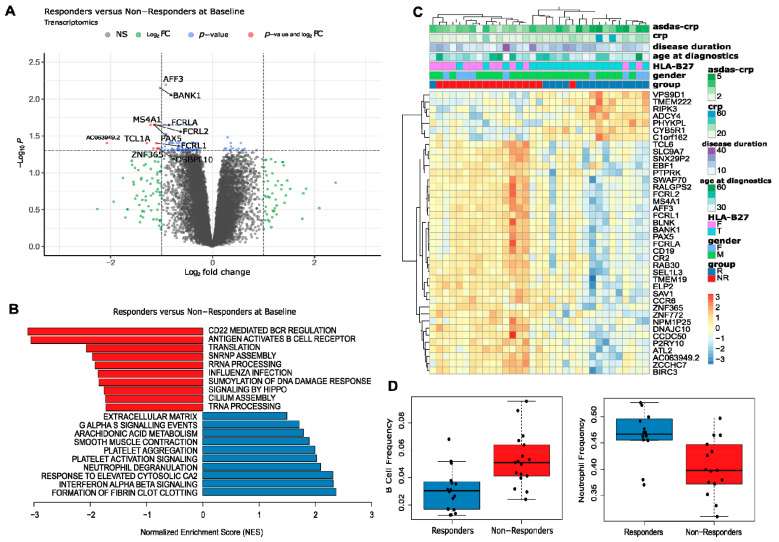
Blood transcriptome data at baseline suggest that the response to adalimumab derives from an interplay between innate and adaptive immunity. (**A**) Volcano plot (log2 of the fold change versus −log10 of the false discovery rate (FDR)) comparing the transcript levels of responder versus non-responder samples at baseline. In grey are non-significant (NS) genes; in green are genes that are not statistically significant (FDR > 0.05) but have an estimated fold change greater than 2; in blue are genes that are statistically significant but have a milder fold change (less than 2); and in red are genes that are statistically significant and have a fold change greater than 2. The names of all the red genes and *AFF3* are displayed in the plot. (**B**) Barplot displaying the Normalized Enrichment Score (NES) of representative significant pathways resulting from a gene set enrichment analysis (GSEA) comparing the gene expression of responder (blue) versus non-responder (red) samples at baseline. (**C**) Heatmap representation of the expression profile of the top 40 differentially expressed genes comparing responder versus non-responder samples at baseline; for visualization purposes, expression values of each gene were scaled towards a standard distribution (z-score), and rows and columns were clustered by correlation. Z-scores of the expression values are presented in a scale from blue (relatively less expressed) to red (relatively more expressed). Sample metadata is presented at the top: ASDAS-CRP score; C-Reactive Protein (mg/L); Disease duration (years); Age at Diagnosis (years); HLA-B27 Status (F—Absent; T—Present); HLA-B27 Status (F—Absent; T—Present); Gender (F—Female; M—Male); Response Group (R—Responders; NR—Non-responders). (**D**) Estimated relative frequencies of B-cell and neutrophil in responder and non-responder samples at baseline.

**Figure 5 biomolecules-14-00382-f005:**
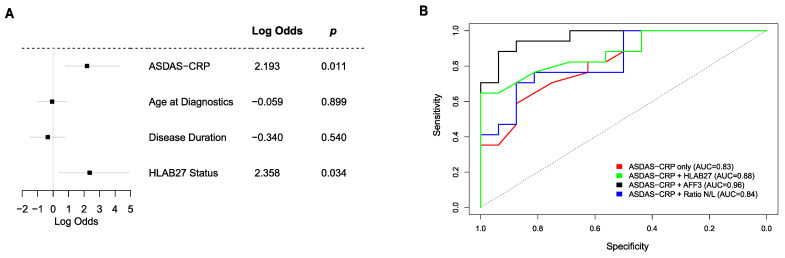
Blood transcriptome data improve the ability to differentiate responders versus non-responders at baseline. (**A**) Forest plot displaying the logarithm of the odds, 95% confidence interval, and *p*-value of response to adalimumab for different variables from a logistic regression model. (**B**) Receiver operating characteristic (ROC) curve displaying specificity and sensitivity depending only on values for ASDAS-CRP, a logistic regression model incorporating ASDAS-CRP, gender, and *HLA-B27* status, a model incorporating ASDAS-CRP, gender, and the expression value of *AFF3*, and a model incorporating ASDAS-CRP, gender, *AFF3*, and *HLA-B27*.

**Table 1 biomolecules-14-00382-t001:** Summary of the clinical characteristics of the cohort. For each continuous variable, the mean and standard deviation within each group were calculated. The two-sample Wilcoxon test (continuous variables) and chi-square test of association (categorical variables) were used to compare characteristics between non-Responders (NR) and responders (R). Variables include Erythrocyte Sedimentation Rate (ESR, in mm/h), C-reactive protein (CRP, in mg/L), Bath Ankylosing Spondylitis Disease Activity Index (BASDAI) scores, Bath Ankylosing Spondylitis Functional Index (BASFI) scores, and Ankylosing Spondylitis Disease Activity Score (ASDAS) using the ESR levels (ASDAS-ESR) or CRP levels (ASDAS-CRP). For these characteristics, the value at baseline and week 14 is provided, as well as the difference of the values between the two endpoints. Other fixed clinical characteristics include age at diagnosis (in years of age), disease duration (in years since start of first symptoms), presence (positive) or absence (negative) of the HLA-B27 allele, and sex (biological gender)—female or male.

	NR (N = 17) Mean (sd)	R (N = 18) Mean (sd)	*p*
**Erythrocyte Sedimentation Rate** **(mm/h)**			
Baseline	26.1 (20.4)	33.2 (28.5)	0.541
Week 14	11.9 (10.7)	10.8 (9.10)	0.856
BL-W14	14.2 (16.3)	22.3 (23.5)	0.298
**C-Reactive Protein** **(mg/L)**			
Baseline	11.3 (11.5)	23.7 (19.7)	0.011
Week 14	7.42 (10.9)	3.90 (2.78)	0.754
BL-W14	3.89 (4.78)	19.8 (19.4)	<0.001
**BASDAI score**			
Baseline	5.35 (2.63)	6.53 (1.46)	0.234
Week 14	4.07 (2.18)	1.93 (1.44)	0.003
BL-W14	1.28 (1.50)	4.60 (1.81)	<0.001
**BASFI score**			
Baseline	5.29 (2.72)	6.71 (1.91)	0.156
Week 14	3.71 (2.65)	2.55 (2.10)	0.176
BL-W14	1.58 (1.49)	4.16 (2.06)	<0.001
**ASDAS-ESR score**			
Baseline	3.23 (0.86)	3.76 (1.08)	0.203
Week 14	2.27 (0.93)	1.48 (0.50)	0.008
BL-W14	0.96 (0.57)	2.28 (1.00)	<0.001
**ASDAS-CRP score**			
Baseline	3.16 (0.75)	4.16 (0.76)	<0.001
Week 14	2.46 (0.75)	1.56 (0.57)	<0.001
BL-W14	0.70 (0.51)	2.59 (0.94)	<0.001
**Age at Diagnosis** **(years)**			
	37.9 (11.3)	34.9 (11.6)	0.301
**Disease duration** **(years)**			
	14.8 (12.7)	13.7 (7.49)	0.869
**HLA-B27 Status**			0.010
Absent	10 (58.8%)	3 (16.7%)	
Present	7 (41.2%)	15 (83.3%)	
**Gender**			0.915
Female	5 (29.4%)	5 (27.8%)	
Male	12 (70.6%)	13 (72.2%)	

## Data Availability

All data are available in the Appendix A. Mass spectrometry data are available through the ProteomeXchange Consortium via the PRIDE partner repository (https://www.ebi.ac.uk/pride/), under dataset identifiers PXD026189 and http://doi.org/10.6019/PXD026189.

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
