# Peer review of "Molecular Profiling of Axial Spondyloarthritis Patients Reveals an Association between Innate and Adaptive Cell Populations and Therapeutic Response to Tumor Necrosis Factor Inhibitors"

_biomolecules, 2024, doi:10.3390/biom14030382_

Round 1

Reviewer 1 Report

Comments and Suggestions for Authors

1.      Line 58: in your brief summary of current treatment options for axSpA please include sulfasalazine and JAK inhibitors.

2.      Line 99: “2.1. Study Design and Samples Collection” – “axSpA patients were selected from participants of the Bioefficacy study”. Reading this phrase the reader understands that there is this “Bioefficacy study” from which the authors extracted their patient sample. Therefore, not all of the patients from the “Bioefficacy study” were included in the study under review. If a part was selected, how was it done? How did the authors decide which “Bioefficacy study” will be included in this study and which not? So please briefly explain sub-sample selection criteria. Was this selection random? How was randomness of inclusion achieved? Please briefly explain this in the manuscript.

3.      Line 99: “2.1. Study Design and Samples Collection” – the mention that the “Portuguese Society of Rheumatology Guidelines” were used probably means that the study took place in Portugal, not sure. But when did the study take place? There is no indication of time frame. It is well established that time and population are important factors for genetic research.

4.      Line 103: “NCT02492217” – does this mother study have anything published (design, results, pilot)? If it does, line 103 would be a very good spot to cite it.

5.      Line 129: “bayes approach” – you probably mean “ Bayes’ ” or even better “Bayesian”.

6.      Table 1: reporting variables with 3 decimals is a bit tiering to read. It does provide credence to your numbers, but it really makes to table hard to follow.

7.      Table 1 mentions BASFI, but the methods section does not.

8.      Table 1: HLA-B27 positive negative. Please be reminded that HLA-B27 is present or absent, since it is a gene.

9.      Reading Table 1 one gets confused: responders have at W14 higher ESR, CRP, BASDAI, BASFI and ASDAS compared to non-responders. Is this correct? Or did you mislabel the collum by switching them?

Author Response

  1. Line 58: in your brief summary of current treatment options for axSpA please include sulfasalazine and JAK inhibitors.

Reply: We modified the introduction accordingly:

" However, biological disease-modifying antirheumatic drugs (bDMARDs) including TNF inhibitors (TNFi), IL-17 and IL-23 inhibitors (IL-17i, IL23i), and more recently, target synthetic DMARDs (tsDMARDs) as JAK inhibitors, are recommended to patients enduring active disease despite conventional treatment (or intolerance / contraindication)[3]." 

  1. Line 99: “2.1. Study Design and Samples Collection” – “axSpA patients were selected from participants of the Bioefficacy study”. Reading this phrase the reader understands that there is this “Bioefficacy study” from which the authors extracted their patient sample. Therefore, not all of the patients from the “Bioefficacy study” were included in the study under review. If a part was selected, how was it done? How did the authors decide which “Bioefficacy study” will be included in this study and which not? So please briefly explain sub-sample selection criteria. Was this selection random? How was randomness of inclusion achieved? Please briefly explain this in the manuscript.

Reply: We thank the reviewer for the question, which led us to notice an inconsistency in the text. We stopped recruitment once we achieved the target number of non-responders (18). As we had more responders, we selected the 18 responders that had better response (and not purely randomly as was previously written). We have modified the text to clarify this:

"Of the 58 patients enrolled in the entire study (40 responders and 18 non-responders), 36 patients (the 18 responders with the best response and all the 18 non-responders) were selected [...]" 

  1. Line 99: “2.1. Study Design and Samples Collection” – the mention that the “Portuguese Society of Rheumatology Guidelines” were used probably means that the study took place in Portugal, not sure. But when did the study take place? There is no indication of time frame. It is well established that time and population are important factors for genetic research.

Reply: We thank the reviewer for the comment. We have modified the methods section to clarify these aspects:

"The Bioefficacy study - Biomarkers Identification of Anti-TNFα Agent's Efficacy in Ankylosing Spondylitis Patients Using a Transcriptome Analysis and Mass Spectrometry (clinical trials.gov identifier NCT02492217), is multicentric, conducted across multiple Rheumatology departments in mainland Portugal. It is a prospective, non-randomized study, spanning 14-weeks focusing on adult patients with axSpA (detailed in table 1), aged between 18 and 75 years old, fulfilling the axSpA ASAS criteria [22]. The study was conducted from 2014 to 2019,  and involved biologic naïve patients initiating TNFi therapy with adalimumab (40mg subcutaneously fortnightly), according to the Portuguese Society of Rheumatology Guidelines[23] "

  1. Line 103: “NCT02492217” – does this mother study have anything published (design, results, pilot)? If it does, line 103 would be a very good spot to cite it.

Reply:  We thank the reviewer for the comment. The design had been submitted, but due to an internal miscommunication, it has not yet been published, although it should be soon. We will see with the editor if it is possible to include.

  1. Line 129: “bayes approach” – you probably mean “ Bayes’ ” or even better “Bayesian”.           

Reply: We have modified to the most commonly used term 'empirical Bayes method'.

  1. Table 1: reporting variables with 3 decimals is a bit tiering to read. It does provide credence to your numbers, but it really makes to table hard to follow.

Reply: We have reduced the number of decimals.

  1. Table 1 mentions BASFI, but the methods section does not.

Reply: Thank you for reporting this omission. We have added BASFI to the methods section:

"Clinical evaluations (Bath Ankylosing Spondylitis Activity Index- BASDAI,  Bath Ankylosing Spondylitis Functional Index- BASFI, Bath Ankylosing Spondylitis Metrology Index- BASMI and 36 – Item Short – Form Health Survey- SF-36) and peripheral blood collections were performed at baseline (BL) (start bDMARD), and after 3-5 days (D3), 2 weeks (W2) and at 14 weeks (W14). "

  1. Table 1: HLA-B27 positive negative. Please be reminded that HLA-B27 is present or absent, since it is a gene.

Reply: The Table was modified to indicate present/absent instead of positive/negative.

  1. Reading Table 1 one gets confused: responders have at W14 higher ESR, CRP, BASDAI, BASFI and ASDAS compared to non-responders. Is this correct? Or did you mislabel the collum by switching them?

Reply: We have changed the legend to make it clearer that the first column in Table1 corresponds to Non-Responders (NR) and the second to Responders (R). As can be seen in Table1, at Week 14 NR have worse (higher) values of ESR, CRP, BASDAI, BASFI and ASDAS compared to R, even though at Baseline NR have on average lower scores in these variables than R.

Reviewer 2 Report

Comments and Suggestions for Authors

A well written study in which the conclusions are supported by data in which The authors demonstrate potentially a means to prospectively define a responder versus non-responder to TNFi therapy for ankylosing spondylitis. I am not as familiar with these statistics but if they are supported by statistician I think the paper is worthy of publication. 

Comments on the Quality of English Language

Except for line 415, there are no problems with syntax or Grammar

Author Response

We thank the reviewer for the positive comments.

We have restructured the paragraph around line 415 to improve the grammar and make it clearer:

"In addition, this work contributes to validate results from previous studies. Nonetheless, we acknowledge limitations in our study, such as the involvement of patients from only one country, the use of only one TNFi and the short period of follow up."